# Comparative Study of Electrospun Polydimethylsiloxane Fibers as a Substitute for Fluorine-Based Polymeric Coatings for Hydrophobic and Icephobic Applications

**DOI:** 10.3390/polym16233386

**Published:** 2024-11-30

**Authors:** Adrián Vicente, Pedro J. Rivero, Cleis Santos, Nadine Rehfeld, Rafael Rodríguez

**Affiliations:** 1Engineering Department, Campus de Arrosadía S/N, Public University of Navarre, 31006 Pamplona, Spain; pedrojose.rivero@unavarra.es (P.J.R.); rafael.rodriguez@unavarra.es (R.R.); 2Institute for Advanced Materials and Mathematics (INAMAT2), Campus de Arrosadía S/N, Public University of Navarre, 31006 Pamplona, Spain; 3Paint Technology Department, Fraunhofer Institute for Manufacturing Technology and Advanced Materials (IFAM), 28359 Bremen, Germany; nadine.rehfeld@ifam.fraunhofer.de; 4Electrical Energy Storage Department, Fraunhofer Institute for Manufacturing Technology and Advanced Materials (IFAM), 28359 Bremen, Germany; cleis.santos@ifam.fraunhofer.de

**Keywords:** electrospinning, PDMS, SLIPS, fibrous mat, ice adhesion

## Abstract

The development of superhydrophobic, waterproof, and breathable membranes, as well as icephobic surfaces, has attracted growing interest. Fluorinated polymers like PTFE or PVDF are highly effective, and previous research by the authors has shown that combining these polymers with electrospinning-induced roughness enhances their hydro- and ice-phobicity. The infusion of these electrospun mats with lubricant oil further improves their icephobic properties, achieving a slippery liquid-infused porous surface (SLIPS). However, their environmental impact has motivated the search for fluorine-free alternatives. This study explores polydimethylsiloxane (PDMS) as an ideal candidate because of its intrinsic properties, such as low surface energy and high flexibility, even at very low temperatures. While some published results have considered this polymer for icephobic applications, in this work, the electrospinning technique has been used for the first time for the fabrication of 95% pure PDMS fibers to obtain hydrophobic porous coatings as well as breathable and waterproof membranes. Moreover, the properties of PDMS made it difficult to process, but these limitations were overcome by adding a very small amount of polyethylene oxide (PEO) followed by a heat treatment process that provides a mat of uniform fibers. The experimental results for the PDMS porous coating confirm a hydrophobic behavior with a water contact angle (WCA) ≈ 118° and roll-off angle (αroll-off) ≈ 55°. In addition, the permeability properties of the fibrous PDMS membrane show a high transmission rate (WVD) ≈ 51.58 g∙m^−2^∙d^−1^, providing breathability and waterproofing. Finally, an ice adhesion centrifuge test showed a low ice adhesion value of 46 kPa. These results highlight the potential of PDMS for effective icephobic and waterproof applications.

## 1. Introduction

Fluorinated polymers are commonly used to create superhydrophobic and waterproof coatings and membranes [1], although their biological persistence, bioaccumulation, and ecotoxicity are key drawbacks concerning their environmental impact [2,3,4]. An interesting polymer, which provides low surface energy without the presence of fluorine, is polydimethylsiloxane (PDMS). This polymer exhibits low glass transition temperature, main chain flexibility, hydrophobicity based on the alkyl side chain, non-toxicity, and non-flammable behavior, and can form elastomers [5,6]. Thus, PDMS is a highly flexible material, even at very low temperatures (−100 °C), suitable for icephobic surfaces [3,7,8]. One interesting technique that can be used for modulating the wetting properties in terms of hydrophobicity or superhydrophobicity is electrospinning [9]. This technique is an electrohydrodynamic process resulting in the movement of a fluid by means of the application of an electrostatic field, making possible the formation of fibers with a tunable diameter (from micron to nanometer) as a function of operational parameters (i.e., flow rate, applied voltage, or distance tip-to-collector) [10,11]. One important point to be highlighted is that PDMS fibers are difficult to obtain by electrospinning due to the high mobility of the chains generated by the low Tg and low molecular weight of the PDMS (low viscosity, limited elasticity in the Taylor cone), lack of polar groups, and dissolution properties. For this reason, in these applications, PDMS is often combined with other carrier polymers to form the final electrospun fiber matrix. Some of these polymers are polystyrene (PS) [12], polyacrylonitrile (PAN) [13], polyamide (PA) [14], polyvinyl alcohol (PVA) [15], polymethylmethacrylate (PMMA) [16], or poly(vinylidene fluoride (PVDF) [17], among others. These types of composite fiber mats, in which the maximal purity reported was 70%, give rise to superhydrophobic coatings or breathable waterproof membranes (WBMs) that can be implemented in different fields where control of the wettability is required.

In addition, these coatings can be an ideal starting point to produce Slippery Liquid Infused Porous Surfaces (SLIPSs) [18], which entrap low surface energy liquids, maintaining both superhydrophobic and icephobic properties and enabling the design of long-term durability with an optimal combination of high water repellency and icephobicity behavior. By using this methodology, the resultant pores on a superhydrophobic surface are infused with a water-immiscible oil (e.g., silicon oil) so that when a water droplet is placed on the surface, the air–liquid contact on a superhydrophobic surface is replaced by a liquid–liquid contact [19]. In the resultant silicon oil-infused PDMS coatings, the low surface energy of both silicon oil and PDMS combined with the high mobility of silicon oil can play an important role in the icephobic property [20,21,22]. The low elastic module of PDMS enables a lower ice adhesion strength, whereas the addition of lubricating low-surface tension oil reveals a greater reduction in ice adhesion [20]. A representative example can be found in [21], where first, a PDMS rough coating was obtained by adding silica nanoparticles, and then silicon oil was successfully infused for the PDMS matrix to produce SLIPSs, obtaining ice adhesion values of around 75 kPa. In the work presented by Yeong et al. [22], low values (<20 kPa) were achieved by replicating microtextures from a laser-irradiated aluminum substrate to an oil-infused PDMS elastomer. To date, very few research works devoted to the combination of PDMS electrospun fibers with SLIPSs [23,24] or the design of highly effective icephobic coatings [25,26,27] can be found in the literature. According to this last topic, the icephobic electrospun SLIPSs are mainly based on fluorine compounds such as poly(vinylidene fluoride-co-hexafluoropropylene) (PVDF-co-HFP) [25,26] or pure polytetrafluoroethylene (PTFE) [27], which have remarkably low ice adhesion values.

Table 1 reports on a selection of representative results (wettability and ice adhesion) on both dry and infused coatings. First, A to F show the cases of fluorine-based electrospun coatings without and with infused silicon oil. Second, the most relevant works that use a PDMS matrix to obtain icephobic coatings are compiled (from G to N), and third, some commercial references used as icephobic surfaces in this work are presented (O and P).

In this work, a high purity (95%) PDMS fibrous coating deposited by the electrospinning technique was studied as a potential substitute for fluorine-based polymeric coatings in hydrophobic and icephobic applications, and it was fully characterized by using SEM, FTIR, and TGA. Further infusion of this coating with silicone oil allowed the creation of SLIPSs. Static and dynamic wettability properties were determined for both the dry and infused coatings. The permeability of the fibrous PDMS membrane was also evaluated to demonstrate that this material and the fabrication strategy proposed here offer an effective solution for controlling moisture and maintaining water resistance in a variety of applications, providing breathability and waterproofing. Finally, the icephobic behavior under static ice conditions was studied in a centrifuge test to confirm the improvement of these novel PDMS surfaces as icephobic coatings and to contrast them with the results summarized in Table 1.

## 2. Experimental Procedure

### 2.1. Materials and Reagents

The membranes were deposited on flat substrates of standard glass slides (76 × 26 mm^2^, Sigma-Aldrich) to identify the characteristics of the thickness, morphology, wettability, and chemical composition, and on spherical test discs (Ø_ext_ of 11.8 cm, Ø_int_ of 10 cm) to measure the permeability properties. The coatings for the ice adhesion test were deposited on AA6061-T6 aluminum flat substrates (dimension 30 × 30 × 3 mm^3^). These substrates were smoothed and cleaned with isopropanol to obtain a roughness below Ra = 0.8 μm prior to the application of the material.

Poly(ethylene oxide) (PEO, (-CH_2_ CH_2_ O-)n, Mw ≈ 5,000,000 g/mol) was acquired from Sigma-Aldrich (Saint Luis, MO, USA). The PDMS emulsion (LIOSIL HC 303 E, PDMS-1) was supplied by IMCD (Cologne, Germany). The aqueous emulsion contained 17 wt% nanoparticles with an average particle size of 25 nm in a non-ionic surfactant, and the lubricant silicon oil ([-Si(CH_3_)_2_ O-]_n_, viscosity 1000 cSt (25 °C)) was acquired from Sigma-Aldrich (Saint Luis, MO, USA).

### 2.2. Fabrication Techniques

#### 2.2.1. Solution and Dispersion Preparation

Firstly, PDMS-1 solid content was increased under vacuum evaporation to 35 wt%, obtaining the PDMS-2 emulsion. Secondly, The PEO was dissolved in a 1:1 ($w_{DMF}:w_{H_{2}O}$) mixture of DMF and distilled water, creating a 2 wt% PEO solution. The solution was prepared at room temperature with vigorous stirring at 700 rpm for 65 h. Subsequently, the PDMS-2 emulsion was incorporated into the mixture at a weight ratio of 95:5 (PDMS solid content/PEO solid content). This process resulted in a uniform PDMS-PEO solution that was maintained at room temperature under gentle stirring at 200 rpm for 12 h.

#### 2.2.2. Electrospinning Setup (ES)

A homemade electrospinning machine installed within a laboratory fume hood was used. It allows the control of the relevant operating parameters such as flow rates, evaporation distance, electrical currents, and voltages. In addition, Taylor Cone visualization provides an accurate fit for these parameters, where a double polarization voltage is applied (needle/collector) to remove the “fly out” nanoparticles and enhance the fiber deposition. The negative electrode was a flat collector, while the positive electrode was a needle with an outer/inner diameter of 0.9/0.6 mm. The process was completed at room temperature (24 ± 3 °C) and 38 ± 5% relative humidity (RH). Table 2 summarizes the operational parameters used in this process.

#### 2.2.3. Coating Fabrication Process

The F(PDMS) and F(SLIPS) coatings were prepared onto the glass slide substrates using a two-step method that included firstly, an electrospinning process, denoted as F(PDMS), and secondly, a further silicon oil infusion, denoted as F(SLIPS). This dual-step manufacturing process is presented in Table 3 and Figure 1.

(i)
*Electrospinning + HT_0_*


In the first step (Figure 1i), the PDMS-PEO solution was loaded into a 10 mL syringe located vertically on the electrospinning system for the preparation of the electrospun membrane with the parameter configurations shown in Table 2 (ES). Then, the sample was placed in a high-temperature muffle at 270 °C (HT_0_) for 60 min to remove PEO, achieving a pure PDMS fibrous structure.

(ii)
*Silicon oil infusion*


Once the first step had been performed, the coating was cooled to room temperature, obtaining the F(PDMS) sample. Then, the sample was cleaned with isopropanol and left to dry for 15 min, preparing the surface for lubricant infusion. To infuse the lubricant, silicone oil was applied drop by drop across the entire surface using a syringe. The sample was then placed on a plate tilted at a 45° angle and left overnight to allow any excess oil to drain off (Figure 1ii), resulting in the F(SLIPS) sample.

### 2.3. Characterization Techniques

Sample thickness was determined using a Mahr Millimar C 1208 micrometer (Mahr GmbH-Göttingen, Germany) at several points, employing an inductive probe with an accuracy of 99.7%. The surface morphology of the sample was analyzed using a field-emission scanning electron microscope (FE-SEM, LEO 1530, Zeiss GmbH, Jena, Germany). FE-SEM images were processed with the ImageJ software to calculate the average fiber diameter (Df) and particle size (Ps) of the F(PDMS) sample. To identify the functional groups present in the samples, Attenuated Total Reflection Fourier-Transform Infrared (ATR-FTIR) spectroscopy analysis was performed. The ATR-FTIR spectra were measured using an Alpha II spectrophotometer (Bruker, Ettlingen, Germany) in the 400–4000 cm^−1^ range.

Wettability tests were performed using the Drop Shape Analyzer DSA 100S (Krüss GmbH, Hamburg, Germany) in accordance with the DIN EN ISO 19403-2 standards. The water contact angle (WCA) was measured by delivering a water droplet at a rate of 0.2 μL/s, with a total volume of 6.0 μL. The sliding angle (WSA) or roll-off angle was determined by tilting the surface at a speed of 60°/min and using a water droplet volume of 20 μL, where the advancing and receding angles of the droplet were measured as it moved at least 1 mm from its initial position [33].

To evaluate the permeability of the membranes, a diffusion of water vapor (WVD) test was carried out using a Labthink C406H tester according to ASTM F1249 and ISO 15106-2. Circular samples with an area of 50 cm^2^ were cut from the coated PET films and clamped into the tester. Table 4 shows the measurement parameters for WVD and OD.

Ice adhesion centrifuge tests were performed at the Fraunhofer IFAM ice lab (Bremen, Germany) using a centrifuge adhesion test (CAT) within a cold climate chamber, ensuring temperature stability throughout the static ice formation process and transport to the CAT. The ambient temperature was constant to the accretion temperature (−8 °C) as well as during the pre-conditioning and adhesion tests. This temperature was selected because it is low enough to prevent unstable icing conditions (which can occur near 0 °C). Test samples were pre-conditioned to this test temperature before the ice formation procedure started. For the assessment of ice adhesion, the static ice type was established according to the icing conditions shown in Table 5.

To create static ice for each test sample, a silicone mold was used to ensure precise ice formation in the required area. Three milliliters of de-ionized water were poured into the mold and allowed to freeze on the test surface (see Figure 2a in [27]). After 90 min, the silicone mold was removed, and the sample was held at the test temperature for a minimum of 15 min without any mechanical disturbance before being placed in the centrifuge. This process resulted in a well-defined layer of compact glaze ice (see Table 4, “Static Ice at IFAM”, in [27]).

As described in [34], the ice adhesion, or the shear stress required to detach the ice from the surface, was related to the rotational speed of the centrifuge rotor when the ice made contact with the centrifuge wall. Four samples for each type of coating were tested consecutively twice, and the average value, along with the standard deviation, was then calculated.

## 3. Results and Discussion

### 3.1. Samples Thickness and Surface Morphology

To analyze the surface morphology and thus determine its structure, the coating thickness as well as fiber diameter of the F(PDMS) samples before and after thermal treatment were measured from the SEM image. The SEM images related to as-fabricated PEO/PDMS fibers before thermal treatment (Figure 2a) show a homogenous and random distribution. After thermal treatment, a network of uniform PDMS fibers composed of PDMS fibers melted together after the ES + HT_0_ procedure, as can be seen in detail in Figure 2b. These facts can be explained by the extremely low amount of PEO in the PEO/PDMS composite fibers and the drying induced by HT_0_.

The results of the structural parameter are summarized in Table 5. As can be appreciated in the histogram in Figure 3, a reduction in the fiber diameter is observed from 2.65 ± 0.34 µm (as-fabricated PDMS-PEO fibers) to 1.56 ± 0.30 µm (after thermal treatment). In both cases, a log-normal distribution shape results in the average values presented in Table 6.

### 3.2. Thermogravimetric Analysis

Thermal analysis of the PDMS-PEO (95:5) composite fibers was conducted to determine the decomposition temperature and the calcination temperature. The TGA result presented in Figure 4 exhibits four weight-loss steps. The first weight loss in the range of 34–380 °C was attributed to moisture and other residual solvents. The second weight loss in the range of 380–470 °C corresponded to PEO decomposition and residual surfactant from PDMS emulsion, while the third one in the range of 470–560 °C corresponded to PDMS decomposition. Finally, above 560 °C, the TGA curve became flat, which implies that the main residue left was carbon.

### 3.3. Chemical Composition

To determine the chemical composition of the sample surface, ATR-FTIR analysis was conducted. The ATR-FTIR spectra of the electrospun F(PDMS) sample, shown in Figure 5, reveal symmetric and asymmetric stretching vibrations at 2962 cm^−1^ and 2906 cm^−1^, respectively, corresponding to the −CH3 groups in Si–CH3 of the pristine PDMS. Absorption peaks at 1412 cm^−1^ and 1257 cm^−1^ are associated with the asymmetric and symmetric deformation of CH3 in Si–CH3, respectively. Additional bands observed at 1008 cm^−1^ represent the symmetric stretching and bending vibrations of Si–O in the Si-O–Si chain. The peak at 785 cm^−1^ corresponds to Si–C stretching in Si–CH3 [35,36]. A small amount of PEO was included in the electrospinning solution to facilitate fiber formation. The typical triplet peak of pure PEO is observed between 1143 cm^−1^ and 1058 cm^−1^ in Figure 5, with the peak at 1094.97 cm^−1^ attributed to the stretching vibration of the ether group C–O–C in PEO [37]. Additionally, pure PEO shows a peak at 2875.61 cm^−1^ from the asymmetric CH2 stretching. It is important to note that the F(PDMS) sample does not show any significant contributions from PEO due to the low amount of PEO incorporated into the fibers and the HT_0_ drying process. Moreover, the F(PDMS) sample exhibits spectra similar to those of the pure PDMS reference sample [36].

### 3.4. Wetting Properties

To assess the wettability properties of the samples, both static and dynamic contact angles were measured. The F(PDMS) sample demonstrated hydrophobic behavior. The low surface energy of PDMS due to a non-polar molecular structure and the presence of hydrophobic methyl groups, along with a hierarchical micro/nanostructure based on microfibers composed of nanoparticles, results in a Cassie–Wenzel mixed state [27,38].

The capillary forces resulting from the air trapped inside the gaps hinder the penetration of the liquid into the surface texture [18,39], enhancing the static and dynamic wetting properties (see Table 7).

On the other hand, the F(PDMS) surface shows excellent chemical stability and compatibility, where its porous structure traps the lubricating liquid. This surface is ideal for the development of SLIPSs, repelling water.

The application of a lubricating film with silicon oil along the F(PDMS) porous surface reduces the adhesion forces by filling the voids and increasing the capillarity in the F(SLIPS) sample. As shown in Table 7, the F(SLIPS) has a slightly lower WCA than the F(PDMS), which is related to the reduction in roughness [11].

In addition, the dynamic wetting properties of both samples were measured through the roll-off angle (α_roll−off_) in the F(PDMS) and the water sliding angle (WSA) in the F(SLIPS). Sample F(SLIPS) had a higher water droplet mobility, resulting in a lower WSA due to the low water interaction in F(SLIPS) with respect to F(PDMS) (see Table 7).

### 3.5. Permeability Properties

To study the permeability properties of the membrane, the WVTR was determined. The F(PDMS) membrane shows a high transmission rate of 51.58 g∙m^−2^∙d^−1^ (WVD) compared with different plastic film references. For example, the WVTR of the F(PDMS) membrane is similar to that of the Amorphous PA film and greater than that of PVC [40,41,42].

The combination of high WVTR and hydrophobicity of the F(PDMS) membrane offers an effective solution for moisture control while maintaining water resistance. This ensures both breathability and waterproofing across various applications such as air and gas filters, water treatment membranes, and protective clothing, among others.

### 3.6. Ice Adhesion Performance

The ice adhesion results, using the CAT method to analyze static ice formation on F(PDMS) and F(SLIPS) samples, show low ice adhesion values (see Table 8). These values confirm the icephobic behavior of PDMS due to its low elastic module or high flexibility at low temperatures. In the case of SLIPSs, the combined effect of the fibrous structure of PDMS with the infusion of silicone oil resulted in a higher ice adhesion value (around 75 kPa) due to the degradation of the coating by the lubricant.

### 3.7. Comparative Study

For the purpose of this comparative study, two main criteria have been followed, as reflected in Figure 6. Firstly, this work and a previously reported study are included based on the strategy of the electrospun fibrous matrix for icephobic coatings (Table 1, A and C). Secondly, some commercial references are used as icephobic surfaces (Table 1, O and P). The coating developed in this study, F(PDMS), combines an electrospun fibrous structure made of PDMS; thus, this comparative study provides insight with respect to its fluorinated counterparts and commercial references.

As can be seen in Figure 6, the ice adhesion of F(PDMS) coatings has low values, between those of F(PTFE) and F(PVDF-HFP). In any case, F(PDMS) shows a clear icephobic behavior in comparison to the PTFE-Tape and Standox commercial references.

Finally, this study demonstrates that F(PDMS) exhibits similar or better performance compared to fluorinated fibrous coatings and shows a similar approach to other developed coatings based on acrylic coatings, such as silanized polyisocyanate curing acrylic resin (PUR C25) [35].

## 4. Conclusions

In the present study, a fluorine-free electrospun polymeric coating was successfully prepared by employing a nearly pure PDMS and adding a very low amount of PEO to a PDMS nanoparticle dispersion, followed by heat treatment (275 °C). On the one hand, the surface morphology and chemical composition of the F(PDMS) samples showed uniform PDMS microfibers with homogeneous density and with the same spectra as the PDMS reference sample. On the other hand, in terms of wettability properties, the F(PDMS) and infused F(SLIPS) samples showed a hydrophobic behavior (WCA ≈ 118°). However, the F(SLIPS) sample had a higher water droplet mobility, resulting in a lower WSA of ≈8° due to low water interaction of F(SLIPS) with respect to F(PDMS). In addition, the permeability properties of the F(PDMS) membrane show a high WVD of ≈51.58 g∙m^−2^∙d^−1^, confirming its breathability as well as waterproofing. Finally, the ice adhesion centrifuge test was carried out for static ice and showed low ice adhesion values of 46 kPa for F(PDMS) and slightly higher values for F(SLIPS), confirming the icephobic behavior of PDMS due to its low elastic module and improving the results of some previously reported fluorine fibrous coatings and commercial references. However, in spite of its higher water droplet mobility, the infused fibrous structure does not show a decrease in ice adhesion compared to the non-infused PDMS matrix coatings.

In future work, additional icing tests will be conducted, focusing on rime and glaze ice formation in an ice wind tunnel. Ice adhesion will also be evaluated by CAT. These efforts aim to further explore and validate the performance of these coatings in different ice formation scenarios.

## Figures and Tables

**Figure 1 polymers-16-03386-f001:**
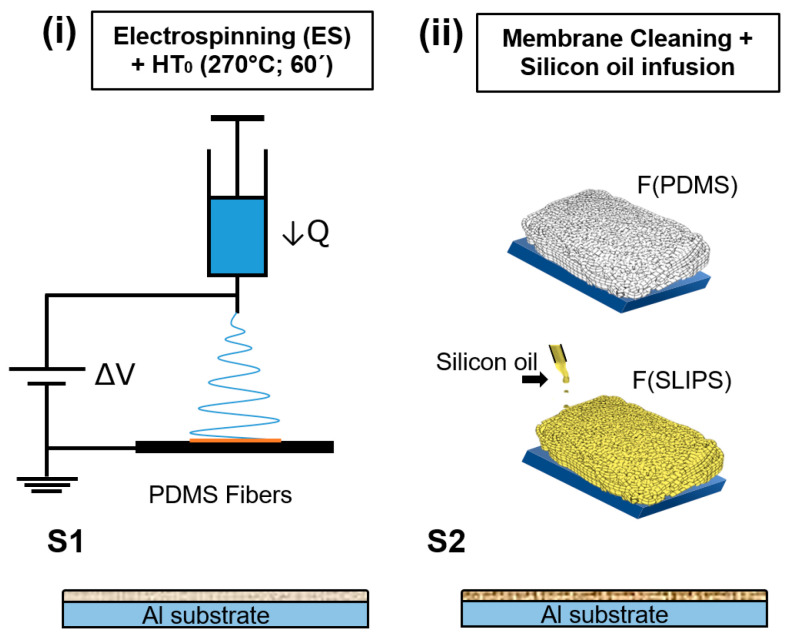
Schematic illustration of the fabrication methods used to produce F(PDMS) and F(SLIPS) samples through the following steps: (**i**) electrospinning corresponding to PEO-PDMS fibrous coating and HT_0_; (**ii**) membrane cleaning + silicon oil infusion.

**Figure 2 polymers-16-03386-f002:**
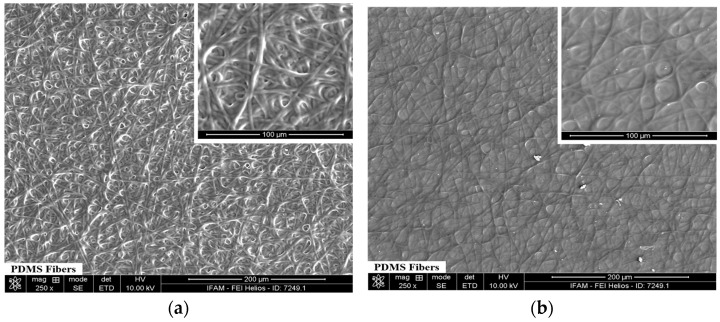
Scanning electron microscopy (SEM) images of the sample surface morphology F(PDMS) before (**a**) and after (**b**) thermal treatment.

**Figure 3 polymers-16-03386-f003:**
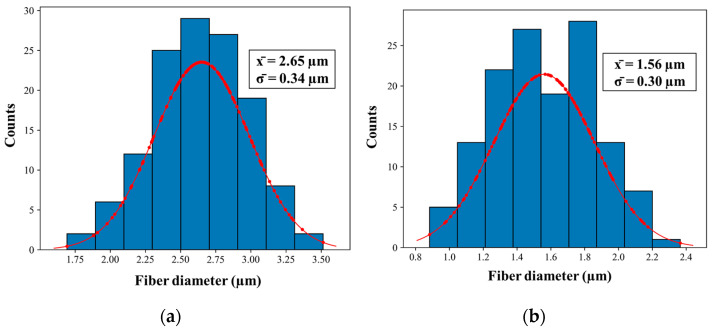
Histograms of the diameter distribution (**a**) and particle size (**b**) of the fiber sample F(PDMS).

**Figure 4 polymers-16-03386-f004:**
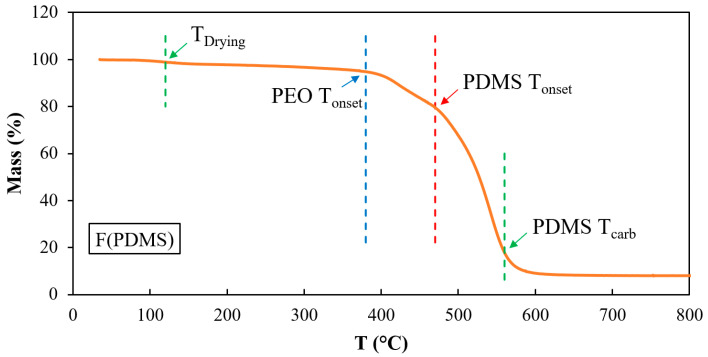
TGA curve of PDMS-PEO composite fibers with a weight ratio of 95:5.

**Figure 5 polymers-16-03386-f005:**
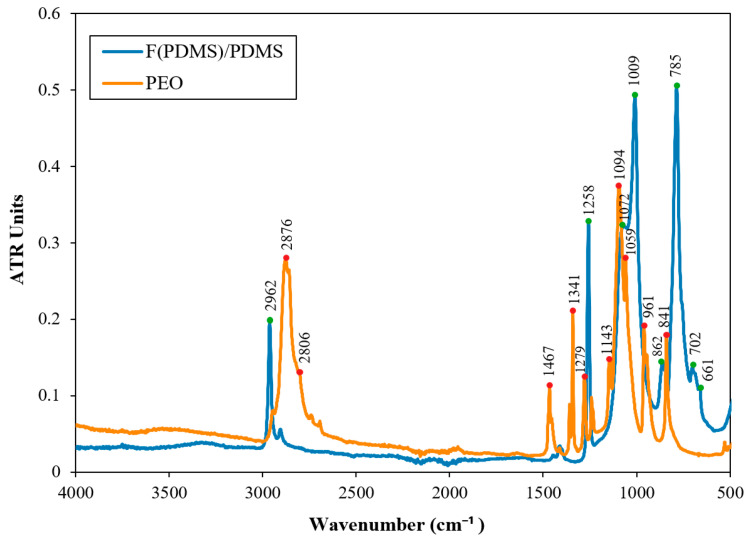
ATR-FTIR spectra of the samples F(PDMS) and PEO.

**Figure 6 polymers-16-03386-f006:**
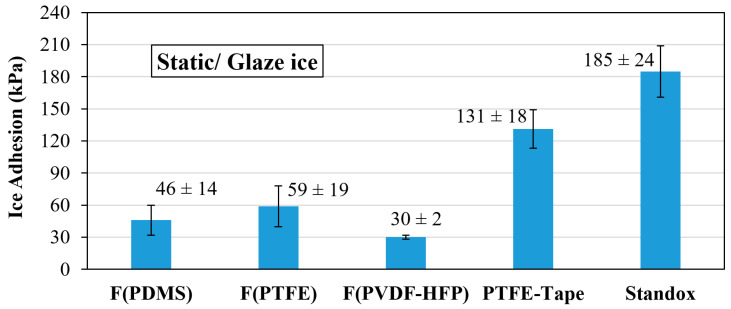
Ice adhesion centrifuge test results for electrospun fibrous icephobic coatings and commercial references based on static ice formation.

**Table 1 polymers-16-03386-t001:** Ice adhesion values of fibrous, infused PDMS-based coating and reference materials.

	Sample	Procedures	WCA	$\alpha_{roll-off}$/ WSA	Icing/Ice Type	Ice Adhesion (kPa)	Ref
A	F(PTFE)	SP +ES + HT	150.3±2.3°	22.8±3.6°	Static/Glaze ^C^	59±19	[27]
B	F(SLIPS-PTFE)	F(PTFE) + Infusion	98.4±0.9°	14.6±1.1°	Static/Glaze ^C^	5±1	[27,28]
					IWT/Rime ^C^	18±12	
					IWT/Glaze ^C^	24±14	
C	F(PVDF-HFP)	ES	141.0±0.7°	―	Static/Glaze ^C^	30±2	[26]
D			161.9±2.4°	50°	IWT/Rime ^C^	89±56	[25]
					IWT/Glaze ^C^	128±16	
E	F(SLIPS-PVDF)	F(PVDF-HFP) + Infusion	106.7 ± 0.5°	27°	IWT/Rime ^C^	11±5	[25]
					IWT/Glaze ^C^	22±8	
F			100.5±0.6°		Static/Glaze ^C^	2±1	[26]
G	PDMS-MNN	Laser ablation + Chem modif.	159°	10°	Static/Glaze *	12.2	[29]
H	TPU-PDMS-10	Solvent vapor annealing + HT	96°	22°	Static/Glaze *	15	[30]
I	PDMS-SO-SiO_2_	Poured + Cured	122°	―	Static/Glaze *	13±3	[31]
J	PDMS MB 10	Cross-linking	113°	―	Static/Glaze *	13±1	[9]
K	PDMS-H_V	Cross-linking	120°	―	Static/Glaze *	75	[21]
L	PDMS 1:2 50% oil	Stirred, degassed, and cured	103.8±3.7°	25°	IWT/Glaze ^C^	9±3	[28]
M			112°	10°	Static/Glaze *	8±1	[20]
N	PDMS-pillars	Cross-linking + cured	165°	3°	Static/Glaze *	26±3	[7]
O	PTFE-Tape	Commercial ref.	110°	29°	Static/Glaze ^C^	131±18	[27,32]
					IWT/Glaze ^C^	102±19	[27]
P	Standox	Commercial ref.	86°	67°	Static/Glaze ^C^	185±24	[32]

* Ice adhesion (Pull off) test; ^C^ Ice adhesion CAT test.

**Table 2 polymers-16-03386-t002:** Configuration of the electrospinning parameters.

Parameters	ES
Applied voltage (needle/collector) (KV)	2.4/−3.2
Flow rate (mL/h)	0.7
Deposition time/sample (min)	30/210
Distance (cm)	26

**Table 3 polymers-16-03386-t003:** Summary of the methods, composition, and heat treatments of each step (Sx).

Acronym	Composition	Procedures	Heat Treatment (HT)		
				Temp (°C)	Time (min)
F(PDMS)	PEO + PDMS	ES + HT_0_	HT_0_	270	60
F(SLIPS)	Si-oil	S1 + Infusion			

**Table 4 polymers-16-03386-t004:** Experimental temperature and relative humidity parameters for diffusion water vapor tests.

Test Gas	H_2_O
Temperature (°C)	38
Relative humidity (%)	90

**Table 5 polymers-16-03386-t005:** Summary of the icing test conditions for static ice.

Temperature (°C)	−8 ± 0.5°
Icing duration	90 min
Ice area (cm^2^)	9
Resulting ice mass (g)	3

**Table 6 polymers-16-03386-t006:** Summary of coating thickness and average fiber diameter of F(PDMS).

Sample	Thickness (µm)	Df (µm), HT_0_ = 25 °C	Df (µm), HT_0_ = 275 °C
F(PDMS)	18 ± 4	2.65 ± 0.34	1.56 ± 0.30

**Table 7 polymers-16-03386-t007:** The water contact angle (WCA), the roll-off water angles (α_roll−off_), and the water sliding angle (WSA) of the F(PDMS) and F(SLIPS) samples.

	F(PDMS)	F(SLIPS)
WCA	118.5 ± 0.9°	103.3 ± 0.3°
α_roll−off_/WSA	55.5 ± 3.6°	8.6 ± 0.6°

**Table 8 polymers-16-03386-t008:** Ice adhesion centrifuge test results for static ice on F(PDMS) and F(SLIPS) samples with their appropriate standard deviations.

Sample	Ice Adhesion (kPa)
F(PDMS)	46±14
F(SLIPS)	79±13

## Data Availability

Data are contained within the article.

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
