# Peer review of "Comparative Study of Electrospun Polydimethylsiloxane Fibers as a Substitute for Fluorine-Based Polymeric Coatings for Hydrophobic and Icephobic Applications"

_polymers, 2024, doi:10.3390/polym16233386_

Round 1
Reviewer 1 Report
Comments and Suggestions for Authors
This manuscript investigates a comparative study of electrospun PDMS fibers as a substitute for fluorine-based polymeric coatings for hydrophobic and icephobic applications. It is a good work. Some aspects should be taken into account.
(1)The main content of the abstract is to extract the essence of the paper, which requires a concise, accurate and comprehensive reflection of the research background, purpose, methods, steps, results, conclusions and other important information of the paper. The abstract is too long and should be concise.
(2)The grammatical tense of the abstract should be consistent.
(3)The electrospinning technique has been used for the first time for the fabrication of 95% pure PDMS fibers to obtain hydrophobic porous coatings as well as breathable and waterproof membranes. Why did the author choose to conduct this research using electrospinning technique?
(4)What are the advantages of the electrospinning technology compared to other fabrication methods?
(5)How to deposit electrospinning PDMS fibers uniformly?This point is very important for the subsequent performance.
(6)For electrospinning parameters, why are these solution parameters selected?
(7)Figure 3 investigates the surface morphology. So, how do surface morphology affect the hydrophobicity of sample?
(8)The low surface energy of PDMS due to a non-polar molecular structure and the presence of hydrophobic methyl groups. Please provide evidence of the existence of functional groups?
(9)For ice adhesion performance. What is the process of obtaining these data? Please provide a detailed description of the experimental process.
(10)What is the relationship between hydrophobicity and ice adhesion?
(11)The latest hydrophobic article with the air–liquid contact replaced by liquid–liquid contact [20] should be cited for great correlations about “The behavior of the interface (changes in phases) affects surface properties” -------"Effect of TiO2 content on the thermal control properties of Al2O3-xTiO2 composite coatings prepared by supersonic plasma spraying technology[J]. Journal of Materials Research and Technology, 2024, 32: 3582-3593."ï¼›
(12)The latest article about “Fluorinated polymers are commonly used to create superhydrophobic and water-proof coatings and membranes” should be cited for great correlations about fluorinated polymers is used to create superhydrophobic and anti-icing behaviors -------"Waterborne robust superhydrophobic PFDTES@TiO2-PU coating with stable corrosion resistance, long-term environmental adaptability, and delayed icing functions on Al-Li alloy[J]. Journal of Materials Research and Technology, 2024, 32: 3357-3370.".
(13)The future work should be explained and added in the end of Conclusion.
(14)The conclusion should be written in point by point.
Author Response
We would like to thank to the anonymous reviewer for his/her positive comments. All the changes have been highlighted in green for a better understanding and localization of them, as it can be appreciated in the revised version of the manuscript. Finally, we hope that this new revised version of the manuscript can be published in Polymers-MDPI.

Reviewer 2 Report
Comments and Suggestions for Authors
The authors addressed 'Comparative study of electrospun PDMS fibers as a substitute for fluorine-based polymeric coatings for hydrophobic and icephobic applications. the paper is interesting. However, some of the points need to be addressed to improve the manuscript
1. Include the novelty statement in the introduction section
2. Properties of silicon oil and deposition on the surface is missing , include it
3. Conclusions should be research outcomes. Please revise the conclusions with results
Author Response

(The authors gave the same response as above.)
